# Domes to Drones: Self-Supervised Active Triangulation for 3D Human Pose Reconstruction

**Aleksis Pirinen,**[1][*] **Erik Gärtner**[1][*] **and Cristian Sminchisescu**[1,2]
[1]Department of Mathematics, Faculty of Engineering, Lund University
[2]Google Research
{aleksis.pirinen, erik.gartner, cristian.sminchisescu}@math.lth.se

## Abstract

Existing state-of-the-art estimation systems can detect 2d poses of multiple people in images quite reliably. In contrast, 3d pose estimation from a single image is ill-posed due to occlusion and depth ambiguities. Assuming access to multiple cameras, or given an *active* system able to position itself to observe the scene from multiple viewpoints, reconstructing 3d pose from 2d measurements becomes well-posed within the framework of standard multi-view geometry. Less clear is what is an informative set of viewpoints for accurate 3d reconstruction, particularly in complex scenes, where people are occluded by others or by scene objects. In order to address the view selection problem in a principled way, we here introduce *ACTOR*, an *active triangulation agent for 3d human pose reconstruction*. Our fully trainable agent consists of a 2d pose estimation network (any of which would work) and a deep reinforcement learning-based policy for camera viewpoint selection. The policy predicts observation viewpoints, the number of which varies adaptively depending on scene content, and the associated images are fed to an underlying pose estimator. Importantly, training the view selection policy requires *no annotations* – given a pre-trained 2d pose estimator, ACTOR is trained in a self-supervised manner. In extensive evaluations on complex multi-people scenes filmed in a Panoptic dome, under multiple viewpoints, we compare our active triangulation agent to strong multi-view baselines, and show that ACTOR produces significantly more accurate 3d pose reconstructions. We also provide a proof-of-concept experiment indicating the potential of connecting our view selection policy to a physical drone observer.

## 1 Introduction

Estimating 2d and 3d human pose from *given* images or video is an active research area, with deep learning playing a prominent role in most of today's state-of-the-art pose and shape estimation models [2, 4, 21–23, 27, 34]. Monocular 3d pose estimation is however ill-posed [26] due to depth ambiguities, and these cannot always be resolved by priors or by increasing a feed-forward model's predictive power. Given access to multiple cameras, or given an *active* observer which can capture images from multiple viewpoints, reconstructing 3d pose from 2d estimates however becomes tractable within the framework of standard multi-view geometry. An active setup for triangulating 2d estimates also addresses many common practical issues, such as partial observability due to occlusion, either self-induced or due to other people or objects.

Given sufficiently many viewpoints, 3d pose reconstructions from 2d estimates can be made robust and accurate, and such results have even been used as (pseudo)ground-truth [17, 33]. While inferring

---
[*]Denotes equal contribution, order determined by coin flip.

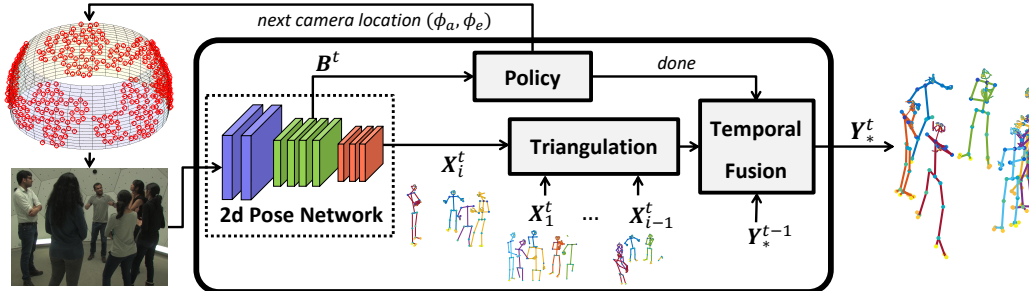

Figure 1: Overview of *ACTOR*, our active triangulation agent for 3d human pose reconstruction. A random view is initially given and the image is fed to a 2d body joint predictor yielding estimates for all visible people ($X_i^t$) and the core of the agent's state ($B^t$). The policy network predicts camera locations until all joints have been triangulated, then switches to the next active-view at time $t+1$. The predicted camera location is encoded via spherical angles relative to the agent's location on the viewing sphere, and the closest camera is selected. When the agent is done it outputs $Y_\star^t$, the final 3d reconstructions of all people in the scene, which are obtained by a combination of spatial fusion (triangulation of 2d poses $X_1^t, \ldots, X_k^t$) and temporal fusion with $Y_\star^{t-1}$ on a per-joint basis. As described in §3.2, we train the viewpoint selection policy using self-supervision.

3d reconstructions from tens or hundreds of viewpoints works in carefully constructed setups, it is not always practical or desirable to rely on so many cameras. In this work we take a different approach, introducing *ACTOR*, an *active triangulation agent for obtaining 3d human pose reconstructions*. ACTOR consists of a 2d pose (human body joints) estimation network (any of which could be used) and a deep reinforcement learning-based policy for observer (i.e. camera location and pose) prediction, within a fully trainable system. Instead of operating exhaustively over all cameras, ACTOR is able to select a much smaller set of cameras yet still produces accurate 3d pose reconstructions. Our proposed methodology is implemented in the Panoptic multi-view framework [17], where the scene can be observed in time-freeze, from a dense set of viewpoints, and over time, providing a proxy for an active observer. In evaluations using Panoptic we show that our system learns to select camera locations that yield more accurate 3d pose reconstructions compared to strong multi-view baselines. We also provide a proof-of-concept experiment indicating the potential of connecting ACTOR to a physical drone observer. Training our policy for view selection requires no 2d or 3d pose annotations – given a pre-trained 2d pose estimator, ACTOR can be trained in a self-supervised manner.

**Related Work.** In addition to recent literature focusing on extracting 3d human representations from a single image or video [3, 21–24, 27, 29, 34, 35], a parallel line of work concentrates on lifting 2d estimates to 3d. [5] present an unsupervised approach for recovering 3d human pose from 2d estimates in single images. This is achieved by a self-consistency loss based on a lift-reproject-lift process, relying on a network that discriminates between real and fake 2d poses in the reprojection step. Related ideas based on an adversarial framework [9] are also pursued in [8]. A self-supervised learning methodology for monocular 3d human pose estimation is described in [19]. During training, the system leverages multi-view 2d pose estimates and epipolar geometry to obtain 3d pose estimates, which are then used to train the monocular 3d pose prediction system. These weakly-supervised methods for monocular 3d pose estimation eliminate the need for expensive 3d ground-truth annotations but tend to not be as accurate as their fully-supervised counterparts.

Multi-view frameworks can, on the other hand, rely on triangulation in order to obtain accurate 3d pose reconstructions given 2d estimates. In contrast to methods performing exhaustive fusion over all cameras, ACTOR actively selects a smaller subset of viewpoints over which to triangulate. Our approach can be considered as a generalization of next-best-view (NBV) selection, and is superficially similar to other NBV-works [11, 12, 14–16, 28, 32]. Differently from them, the number of viewpoints explored by our agent varies adaptively based on the complexity of the scene. Also, NBV approaches typically decide the next view by greedily and locally evaluating some hand-crafted utility function exhaustively over a set of candidates – we instead frame the task as a deep RL problem where the policy is trained to maximize an explicit *global objective*, searching over entire sequences of viewpoints, and by triangulating as many joints as possible. In a broader sense, ACTOR relates to work on active agents trained to perform various tasks in 3d environments [1, 6, 7, 10, 31, 36]. We are not aware of any prior work that tackles the problem of active triangulation in multi-view setups.

## 2 Human Pose Reconstruction from Active Triangulation

We here describe the terminology and concepts of 3d human pose reconstruction from active triangulation. The proposed framework is applicable to any number of people as we aim for a system able to actively reconstruct *all* people in the scene, the number of which may vary. We study the active triangulation problem in the CMU Panoptic multi-camera framework [17] since its data consists of real videos of people and allows for reproducible experiments. The subjects are filmed by densely positioned time-synchronized HD-cameras as they perform movements ranging from basic pose demonstrations to different social interactions. Panoptic offers 2d and 3d joint annotations, but as we will show no such annotations are required for training our viewpoint selection system. See Fig. 1 for an overview of our active 3d human pose reconstruction model.

**Terminology.** Triangulation of 3d pose reconstructions from 2d estimates requires observing the targets from several cameras, each capturing an image $v_i^t$ (referred to as a *view* or *viewpoint*) indexed by time-step $t$ and camera $i$. The set $\{v_1^t, \ldots, v_N^t\}$ of all views in a time-step $t$ is called a *time-freeze*. A subset of these is an *active-view*, $\mathcal{V}^t = \{v_1^t, \ldots, v_k^t\}$, which contains $k$ cameras selected (by some agent or heuristic) from the time-freeze at time $t$. A sequence of temporally contiguous active-views is referred to as an *active-sequence*, $\mathcal{S}^{1:T} = \{\mathcal{V}^1, \mathcal{V}^2, \ldots, \mathcal{V}^T\}$, where $T$ is its length. Unless the context requires both indices we will omit the time super-script $t$ to simplify notation, which implies that all elements belong to the same timestep. The set of all predicted 2d pose estimates corresponding to a view $v_i$ is denoted $\boldsymbol{X}_i = [\boldsymbol{x}_1, \ldots, \boldsymbol{x}_M] \in \mathbb{R}^{30 \times M}$, where $\boldsymbol{x}$ is a single 2d pose estimate, based on detecting 15 human body joints, and $M$ is the number of people observed from that viewpoint.

**Task description.** *Active triangulation for 3d human pose reconstruction* is the task of producing active-views with corresponding accurate fused 3d pose reconstructions for all people present, $\boldsymbol{Y}_\star = [\boldsymbol{y}_{1\star}, \ldots, \boldsymbol{y}_{M\star}]$, given 2d pose estimates $\boldsymbol{X}_1, \ldots, \boldsymbol{X}_k$ associated with the active-view. These active-views then form an active-sequence of accurate 3d pose reconstructions. As it is challenging to select appropriate viewpoints for satisfactory triangulation, especially in crowded scenes where people are often occluding each other, the task is considered completed once each individual's joint has been observed from at least two different viewpoints (the minimum requirement for performing a triangulation), or after a given exploration budget is exceeded.

**Matching and triangulating people.** The active triangulation system must tackle the problems of tracking and identifying people across various views and through time. The agent receives appearance models[2] for the different people at the beginning of an active-sequence. For each view, the agent compares the people detected by the 2d pose estimator with the given appearance models and matches them across space and time using the Hungarian algorithm. To reconstruct 3d poses from 2d estimates associated with the selected viewpoints, we compute triangulation between each pair of viewpoints [13,20] and perform per-body-joint fusion (averaging) of the associated 3d reconstructions. More sophisticated triangulation methods would be possible; here we selected pairwise averaging due to computational efficiency which is important during training.

## 3 Active Triangulation Agent

We now introduce our active triangulation agent, ACTOR, and describe its state representation and action space in §3.1. In §3.2 we describe the *annotation-free* reward signal for training ACTOR to efficiently triangulate the joints of all people.

In the first active-view $\mathcal{V}^1$, the agent is given an initial random view $v_1^1$. It then predicts camera locations $v_2^1, \ldots, v_k^1$ until the active-view is completed. An active-view is considered completed once the agent has triangulated the joints of all people within the time-freeze, or after a given exploration budget has been exceeded. The 2d pose estimator is computed for images collected at every visited viewpoint $v_i^t$, yielding estimates $\boldsymbol{X}_i^t$ for all visible people. Camera locations are specified by the relative azimuth and elevation angles (jointly referred to as *spherical angles*) on the viewing sphere.

Once the agent has triangulated the joints of all people within a time-freeze, it continues to the next active-view $\mathcal{V}^{t+1}$. At this time the triangulated 3d pose reconstructions $\boldsymbol{Y}^t$ are temporally fused with the reconstructions $\boldsymbol{Y}_\star^{t-1}$ from the previous active-view, $\boldsymbol{Y}_\star^t = f(\boldsymbol{Y}_\star^{t-1}, \boldsymbol{Y}^t)$. As the

2d pose estimator we use in this work is accurate, we have opted for a straightforward temporal fusion. We define $I = I_{\text{tri}} \cup I_{\text{miss}}$, where $I$ indexes all joints, $I_{\text{tri}}$ indexes the successfully triangulated joints in the current time-step, and $I_{\text{miss}}$ indexes joints missed in the current time-step. Then we set $\boldsymbol{Y}_\star^t[I_{\text{tri}}] = \boldsymbol{Y}^t[I_{\text{tri}}]$ for the joints that were successfully triangulated in the current time-step, and $\boldsymbol{Y}_\star^t[I_{\text{miss}}] = \boldsymbol{Y}_\star^{t-1}[I_{\text{miss}}]$. Hence we temporally propagate from the previous time-step only those joint reconstructions that were missed in the current time-step. The initial viewpoint $v_1^{t+1}$ for $\mathcal{V}^{t+1}$ is set to the final viewpoint $v_k^t$ of $\mathcal{V}^t$, i.e. $v_1^{t+1} = v_k^t$. The process repeats until the end of the active-sequence. Fig. 1 shows a schematic overview of ACTOR.

## 3.1 State-Action Representation

In this section, while describing the state and action representations, we will assume that the agent acts in a single time-freeze. This allows us to simplify notation and index steps within the active-view by $t$. The state is represented as a tuple $S^t = (\boldsymbol{B}^t, \boldsymbol{C}^t, \boldsymbol{u}^t)$, where $\boldsymbol{B}^t \in \mathbb{R}^{H \times W \times C}$ is the deep feature map from the 2d pose estimator. $\boldsymbol{C}^t \in \mathbb{N}^{w \times h \times 2}$ is a *camera history*, which encodes[3] the previously visited cameras on the rig. It also contains a representation of the distribution of cameras on the rig. The auxiliary array $\boldsymbol{u}^t \in \mathbb{R}^{17}$ contains the number of actions taken, the number of people detected, as well as a binary vector indicating which joints have been triangulated for all people.

A deep stochastic policy $\pi_{\boldsymbol{\theta}}(\boldsymbol{c}^t | S^t)$ parametrized by $\boldsymbol{\theta}$ is used to predict the next camera location $\boldsymbol{c}^t = (\phi_a^t, \phi_e^t)$, were $(\phi_a^t, \phi_e^t)$ is the azimuth-elevation angle pair encoding the camera location. To estimate the camera location probability density, the base feature map $\boldsymbol{B}^t$ is processed through two convolutional blocks. The output of the second convolutional block is concatenated with $\boldsymbol{C}^t$ and $\boldsymbol{u}^t$ and fed to the policy head, consisting of 3 fully-connected layers with $\tanh$ activations.

As the policy predicts spherical angles, we choose to sample these from the periodical von Mises distribution. We use individual distributions in the azimuth and elevation directions. The probability density function for the azimuth angle is given by

$$\pi_{\boldsymbol{\theta}}\left(\phi_a^t | S^t\right) = \frac{1}{2\pi I_0(m_a)} \exp\{m_a \cos(\phi_a^t - \tilde{\phi}_a(\boldsymbol{w}_a^\top \boldsymbol{z}_a^t + b_a))\} \tag{1}$$

where the zeroth-order Bessel function $I_0$ normalizes (1) to a probability distribution on the unit circle. Here $\tilde{\phi}^a$ is the mean of the distribution (parameterized by the deep network), $m_a$ is the precision parameter,[4] and $\boldsymbol{w}_a$ and $b_a$ are trainable weights and bias, respectively. The second to last layer of the policy head outputs $\boldsymbol{z}_a^t$. For the azimuth prediction, the support is the full circle. Therefore we set

$$\tilde{\phi}_a(\boldsymbol{w}_a^\top \boldsymbol{z}_a^t + b_a) = \pi \tanh(\boldsymbol{w}_a^\top \boldsymbol{z}_a^t + b_a) \tag{2}$$

The probability density for the elevation prediction has the same form (1) as the azimuth. As there are no cameras below the ground-plane of the rig, nor cameras directly above the people (cf. Fig. 1), we limit the elevation angle range to $[-\kappa, \kappa]$, where $\kappa = \pi/6$. Thus the mean elevation angle becomes

$$\tilde{\phi}_e(\boldsymbol{w}_e^\top \boldsymbol{z}_e^t + b_e) = \kappa \tanh(\boldsymbol{w}_e^\top \boldsymbol{z}_e^t + b_e) \tag{3}$$

## 3.2 Reward Signal for Self-Supervised Active Triangulation

As explained in §3, ACTOR predicts camera locations until the individual body joints of all people have been detected from at least two different views (minimum requirement for 3d triangulation) or after a given exploration budget $B$ is exceeded; we set $B = 10$ during training. We use the indicator variable $d_t$ to denote whether or not the agent has triangulated all joints ($d_t = 1$ if all joints have been triangulated). We want to encourage the agent to fulfill the task while selecting as few camera locations as possible, which gives rise to the reward design in (4) below. Note that our reward is *not* based on ground-truth pose annotations – it relies solely on automatic 2d pose (body joint) detections.

$$r^t = \begin{cases} -\beta/M, & \text{if } d_t = 0, t < B \text{ and camera not already visited} \\ -\beta/M - \epsilon, & \text{if } d_t = 0, t < B \text{ and camera already visited} \\ 1, & \text{if } d_t = 1, t \le B \\ \tau_{min}, & \text{if } d_t = 0, t = B \end{cases} \tag{4}$$

The first and second rows of (4) reflect intermediate rewards, where the agent receives a penalty $\epsilon$ (we set $\epsilon = 2.5$) if it predicts a previous camera location. To encourage efficiency the agent also receives a time-step penalty $\beta$ for not yet having completed the triangulation ($\beta$ is set to 0.2). This penalty is normalized by the number of people $M$ for scaling purposes, as we expect more cameras be required to triangulate multiple people. The third and fourth rows represent rewards the agent obtains at the end of the active-view. It receives $+1$ if it triangulates the joints of all $M$ persons within its exploration budget $B$. The fourth row defines the reward if the agent fails to triangulate some joints within the exploration budget. It then receives the minimum fraction of covered joints for any person, $\tau_{min}$. Policy gradients are used to learn ACTOR's policy parameters, where we maximize expected cumulative reward on the training set with the objective $J(\boldsymbol{\theta}) = \mathbb{E}_{\boldsymbol{s} \sim \pi_{\boldsymbol{\theta}}} \left[ \sum_{t=1}^{|\boldsymbol{s}|} r^t \right]$, where $\boldsymbol{s}$ denotes state-action trajectories. This objective function is approximated using REINFORCE [30].

| Model | Data | Auto | 2 | 3 | 4 | 5 | 6 | 7 | 8 | 9 | 10 |
|---|---|---|---|---|---|---|---|---|---|---|---|
| ACTOR | multi | 125.6 (8.84) | 502.4 | 281.5 | 201.0 | 168.4 | 151.6 | 141.2 | 132.1 | 126.1 | 122.1 |
| | | 96.2 (8.84) | 247.2 | 179.3 | 146.4 | 131.1 | 118.5 | 111.6 | 101.9 | 95.2 | 92.3 |
| | single | 74.6 (4.28) | 172.1 | 107.5 | 81.9 | 71.2 | 67.1 | 64.9 | 63.3 | 62.1 | 61.3 |
| | | 60.5 (4.28) | 151.3 | 92.8 | 68.9 | 59.4 | 55.6 | 53.2 | 51.3 | 49.9 | 49.0 |
| ACTOR-ob | multi | 148.9 (8.79) | 555.2 | 372.9 | 276.4 | 217.4 | 185.2 | 166.6 | 154.0 | 146.1 | 142.5 |
| | | 108.1 (8.79) | 299.6 | 305.7 | 231.2 | 182.4 | 155.9 | 131.9 | 119.4 | 112.3 | 109.3 |
| | single | 80.2 (4.58) | 187.3 | 122.6 | 95.1 | 80.6 | 72.4 | 68.9 | 67.4 | 67.0 | 66.8 |
| | | 67.3 (4.58) | 159.7 | 104.4 | 77.7 | 64.2 | 56.6 | 53.3 | 52.3 | 51.8 | 51.6 |
| ACTOR-ntf | multi | 138.9 (8.84) | 925.7 | 565.2 | 353.1 | 242.8 | 196.5 | 172.2 | 154.8 | 143.7 | 136.6 |
| | | 102.0 (8.84) | 334.4 | 258.0 | 198.4 | 159.1 | 138.4 | 124.5 | 112.0 | 102.9 | 98.3 |
| | single | 75.9 (4.28) | 274.0 | 151.4 | 99.6 | 79.3 | 71.8 | 67.9 | 65.5 | 63.9 | 62.7 |
| | | 61.6 (4.28) | 228.1 | 132.4 | 85.3 | 66.9 | 59.8 | 55.9 | 53.3 | 51.5 | 50.3 |
| Random | multi | 142.7 (9.34) | 570.1 | 469.9 | 316.1 | 259.9 | 269.3 | 238.5 | 220.2 | 198.8 | 188.3 |
| | | 125.9 (9.34) | 347.3 | 406.4 | 350.1 | 278.0 | 263.0 | 218.8 | 196.2 | 179.5 | 160.0 |
| | single | 82.6 (4.90) | 203.6 | 139.4 | 107.2 | 89.9 | 81.1 | 75.1 | 71.0 | 67.9 | 65.8 |
| | | 68.7 (4.90) | 178.0 | 125.7 | 93.8 | 76.4 | 67.6 | 61.3 | 56.8 | 53.4 | 51.0 |
| Max-Azim | multi | 132.0 (9.01) | 479.3 | 375.8 | 288.4 | 226.0 | 195.7 | 170.2 | 149.2 | 137.7 | 128.6 |
| | | 102.7 (9.01) | 259.4 | 282.1 | 235.0 | 200.0 | 196.8 | 158.2 | 131.3 | 114.1 | 103.7 |
| | single | 75.5 (4.41) | 185.7 | 119.5 | 88.0 | 79.5 | 73.7 | 68.8 | 64.5 | 63.2 | 62.1 |
| | | 63.61 (4.41) | 161.2 | 106.3 | 76.5 | 67.7 | 61.0 | 56.3 | 52.0 | 50.0 | 48.5 |
| Oracle | multi | 94.5 (6.67) | 254.4 | 147.6 | 113.1 | 98.2 | 90.3 | 86.4 | 84.1 | 82.8 | 81.9 |
| | | 74.0 (6.67) | 163.1 | 110.3 | 89.2 | 78.8 | 72.8 | 69.0 | 66.4 | 64.5 | 63.0 |
| | single | 54.0 (2.97) | 123.0 | 60.2 | 49.2 | 45.3 | 43.6 | 42.8 | 42.3 | 42.2 | 42.4 |
| | | 48.1 (2.97) | 108.2 | 54.5 | 43.3 | 39.5 | 37.5 | 36.2 | 35.2 | 34.6 | 34.2 |

Table 1: Mean 3d reconstruction error (mm/joint) for ACTOR and baselines on the Panoptic test sets. *Multi* denotes multi-people data (union of *Mafia* and *Ultimatum*); *single* is the single-person *Pose* split. We show total errors which include translation errors (top) and hip-aligned errors (bottom). Columns indicate the number of cameras inspected, ranging from 2 to 10. We also show results for auto-mode, where camera location selection terminates when the joints of all people have been triangulated, but using 10 cameras at most. For this column we also show the average number of cameras inspected in parentheses. ACTOR outperforms both the heuristic baselines on all types of scenes. The advantage of a trained system is most pronounced for complex multi-people scenes where selecting informative viewpoints is important. ACTOR-ob and ACTOR-ntf denote ablated versions of our agent, cf. §4.2.

## 4 Experiments

**Dataset.** We consider both multi-people scenes (named *Mafia* and *Ultimatum* in Panoptic) and single-people ones (*Pose*). The scenes with multiple people are expected to be particularly challenging for the agent, as occlusions are common. Panoptic data comes as 30 FPS time-synchronized videos. To make the size more manageable and increase movement between frames we downsample the data to 2 FPS. We use the HD cameras, of which there are about 30 per scene, since they provide better image quality than VGA and are sufficiently dense, yet spread apart far enough to make each viewpoint unique. We select 20 scenes (343k images) which are split randomly into training, validation and

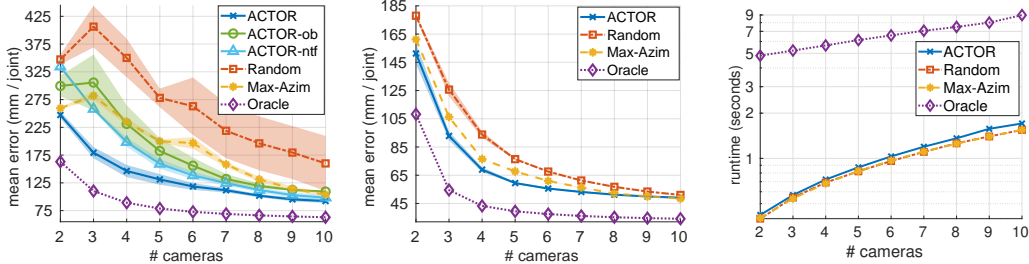

Figure 2: Column 1-2: Mean 3d reconstruction error per joint vs number of cameras on the test sets (means and standard errors over 5 seeds). Column 1: Multi-people data. Column 2: Single-people data. ACTOR decreases errors faster than baselines, particularly for multi-people data with occlusions. The oracle uses 3d ground-truth and is shown as gold standard. ACTOR also outperforms the ablated variants ACTOR-ob and ACTOR-ntf, cf. §4.2. Ablated models are not plotted in the single-people setting to avoid visual clutter – see Table 1 for these results. Column 3: Runtime (log-scale) per active-view vs number of cameras for a 3-people scene. The oracle computes errors using 3d ground-truth for all views and persons to select its next camera, making it very slow.

test sets with 10, 4, and 6 scenes, respectively. There is no overlap of scenes between the three sets, which forces the agent to learn a fairly general policy.

**Implementation details.** ACTOR is implemented on top of the OpenPose 2d pose estimation system [4], though any 2d pose predictor would work. As described in §3, temporal fusion of 3d reconstructions across active-views ensures that missed joints are instead covered by the associated estimates from an earlier point in time. In case there is no previous estimate for a missing joint, it is set to the average of the successfully triangulated ones (to be able to compute errors). We use per-joint median averaging for fusing 3d pose reconstructions across views and temporal steps.

**Training.** We train the policy network with batches consisting of experiences from 5 active-sequences, each of length 10. Adam [18] is used for parameter updates. We normalize cumulative rewards for each episode to zero mean and unit variance over each batch to reduce variance in the policy updates. The exploration budget $B$ (maximum trajectory length) is set to 10 camera locations per active-view, including the initial camera. The policy is trained for 75k episodes with learning rate initially set to $\mathtt{5e-7}$, then halved after 720k steps and again after 1440k steps. The precision parameters $(m_a, m_e)$ of the von Mises distributions are linearly annealed from $(1, 10)$ to $(25, 50)$ during training, which makes the camera prediction increasingly deterministic as the training progresses.

**Baselines.** We evaluate ACTOR against several multi-view baselines. They use the same 2d pose estimator, matching algorithm, triangulation method and temporal fusion. All methods receive the same initial random camera at the start of an active-sequence. We compare to the following baselines: i) *Random:* Selects random cameras (it never selects the same camera twice); *Max-Azim:* The first three views are selected at 90, 180 and 270 degrees azimuth relative to the initial view, so the four first views are at 90 degrees azimuth from each other. The subsequent four views are also selected at 90 degrees azimuth from each other, but at a 45 degree azimuth offset relative to the first four views. At each azimuth, it samples a random elevation angle. The last 2 cameras are selected randomly, and we ensure each camera is different. This baseline produces a wide coverage of the viewing sphere without the need to know in advance how many cameras will be selected; iii) *Oracle:* Before selecting one camera, this computes the improvement in 3d pose reconstruction error associated with all available cameras. It then selects the camera that maximally decreases the error. In addition to cheating when it selects views, the oracle is also impractically slow since it exhaustively computes errors for all cameras in each step. Thus it is only shown as a gold standard.

## 4.1 Main Results

Our ACTOR agent is compared to the baselines on the Panoptic test sets on active-sequences consisting of 10 active-views. We train ACTOR with 5 different random network initializations and report average results with standard errors of the means (we early stop training for each network initialization based on errors on the validation set). For the non-deterministic heuristic baselines (*Random* and *Max-Azim* – the oracle is deterministic) we report results across 5 seeds, including standard errors of the means. In Table 1 we report 3d pose reconstruction errors for auto stopping and

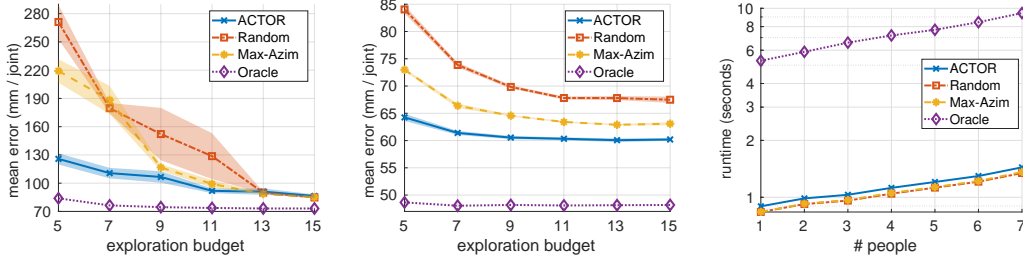

Figure 3: Column 1-2: Mean 3d reconstruction error per joint vs exploration budget $B$ (maximum number of cameras) on the test sets. As mentioned in §3.2, ACTOR was trained solely at budget $B = 10$. Column 1: Multi-people data. Column 2: Single-people data. The relative gain to the baselines is higher at smaller exploration budgets (e.g. 93 mm/joint improvement over *Max-Azim* on multi-people data at $B = 5$), where the system quickly needs to select cameras triangulating the joints of all people. The accuracy curves are flatter for single-people data, as in general the systems need fewer cameras to triangulate the joints of a single person – the models hence tend to stop before their budget is exhausted. Column 3: Runtime (log-scale) when varying the number of people in a scene while keeping the the number of selected cameras constant at 6 per active-view.

for a fixed number of views. ACTOR is more accurate and uses fewer cameras on average, compared to the heuristic baselines. Fig. 2 shows 3d pose reconstruction error versus number of views. ACTOR significantly outperforms the heuristic baselines, especially for complex multi-person scenes (e.g. 103 and 78 mm/joint improvements over *Max-Azim* at 3 and 6 cameras, respectively). Multi-people scenes are more difficult to analyze due to occlusions and thus require intelligent viewpoint selection – one clearly sees the advantages of a learned system in such scenarios.

Fig. 3 shows how the exploration budget $B$ (max number of views) affects 3d reconstruction error. At smaller budgets ACTOR's improvements over the heuristic baselines are even larger, which shows that our trained system is significantly more efficient at finding good views over which to triangulate the body joints. Runtimes versus number of cameras are shown in column 3 of Fig. 2 and versus number of people in column 3 of Fig. 3. OpenPose runs at about $0.134$ seconds per image, while the policy network inference has an overhead of 0.005 seconds per action, which is negligible compared to the 2d pose estimator. For visualizations[5] of ACTOR operating in various scenes, see Fig. 5.

## 4.2 Ablation Studies

In this section we study how ACTOR is affected by i) removing all state features except the deep feature blob $B^t$ (ACTOR-ob; *ob* stands for *only blob*), and ii) using no temporal fusion of 3d pose reconstructions (ACTOR-ntf). Similarly as for the main ACTOR model, ACTOR-ob is trained over 5 different network initializations with individual early stopping on the validation set (ACTOR-ntf uses the same parameters as ACTOR but without temporal fusion during inference). The results are shown in Table 1 and Fig. 2. The full ACTOR agent outperforms the ablated variants for all data splits. For multi-people data, ACTOR drastically outperforms ACTOR-ob, which indicates the need of representing earlier visited cameras ($C^t$) as part of the state space. For single-people data, ACTOR-ob is almost as good as ACTOR, but this data is very simple and occlusion-free and does not require too sophisticated camera selection. Finally, the full agent outperforms ACTOR-ntf when operating using few cameras, which makes sense as there is a big risk of the system missing to triangulate some joints, in which case a backup from earlier active-views may help.

## 4.3 From Domes to Drones

The dense Panoptic multi-camera dome provides an idealization in which we can generate controllable and reproducible experiments. It is also useful for training ACTOR, as we do not actually have to move a camera around. However, in many practical scenarios one does not have access to a multi-view setup and may instead have to resort to a single but moving camera. One such scenario is that of a drone circling a set of people, and aiming to reconstruct their 3d poses.

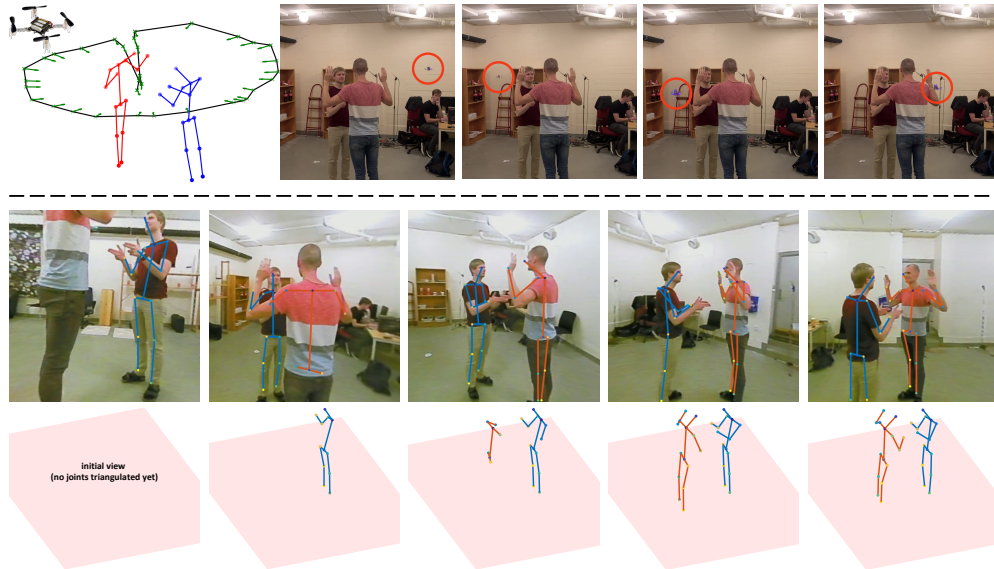

Figure 4: **From domes to drones.** Proof-of-concept experiment illustrating that ACTOR can be connected to an active drone observer to reconstruct 3d poses from informative viewpoints. Above the dashed line to the left we show the drone's loop (the sharp peak is due to take-off and landing), with sampled camera locations as green arrows. We also show the 3d pose reconstructions obtained by triangulating from *all* 33 sampled camera locations. The 9-by-9 cm Crazyflie drone used is shown in the very top left corner; it can be used safely due to its small size and weight. Sample locations of the drone are also shown above the line (drone locations are highlighted with red circles in images). Below the line we show views seen by ACTOR and aggregated 3d pose reconstructions. After observing 5 viewpoints, the two bodies are fully 3d reconstructed, with an average 2d reprojection error of 11.5 pixels (averaged over all 33 cameras), significantly better than the exhaustively triangulated reconstructions to the left, with an average reprojection error of 35.4 pixels.

To test ACTOR's drone-controlling capacity, we captured three small scenes where a drone circles around two people performing various poses. We then fine-tuned ACTOR with learning rate `1e-6` for 3k episodes (15 minutes) on two scenes, keeping all other hyperparameters the same, and ran the model on the third scene. In Fig. 4, ACTOR selects 5 different views to reconstruct the targets. It should be noted that the setting of this drone experiment differs drastically from that of Panoptic. For example, the drone's camera quality is worse (VGA rather than HD), and the loop generated by the drone has a much smaller radius than Panoptic's viewing sphere (about 1.5 meters for the drone versus about 3 meters for Panoptic), so there are fewer views where e.g. the feet are visible. In future work we plan to more tightly integrate ACTOR in the loop, so as to direct the drone to observe targets from informative views.

## 5   Conclusions

We have presented *ACTOR*, a deep RL-based agent to actively reconstruct 3d poses from 2d estimates via triangulation. Training the viewpoint selection policy requires no annotations and only uses an off-the-shelf 2d human pose estimator for self-supervision. We evaluated the model in complex scenarios with multiple interacting people and showed that by intelligently selecting informative views the agent outperforms strong multi-view baselines in both speed and accuracy. We also provided proof-of-concept results which indicate that ACTOR can be used in single-camera settings, e.g. to control a physical drone observer.

**Acknowledgments:** This work was supported by the European Research Council Consolidator grant SEED, CNCS-UEFISCDI PN-III-P4-ID-PCE-2016-0535 and PCCF-2016-0180, the EU Horizon 2020 Grant DE-ENIGMA, Swedish Foundation for Strategic Research (SSF) Smart Systems Program, as well as the Wallenberg AI, Autonomous Systems and Software Program (WASP) funded by the Knut and Alice Wallenberg Foundation. We would also like to thank Patrik Persson for support with the drone experiments.

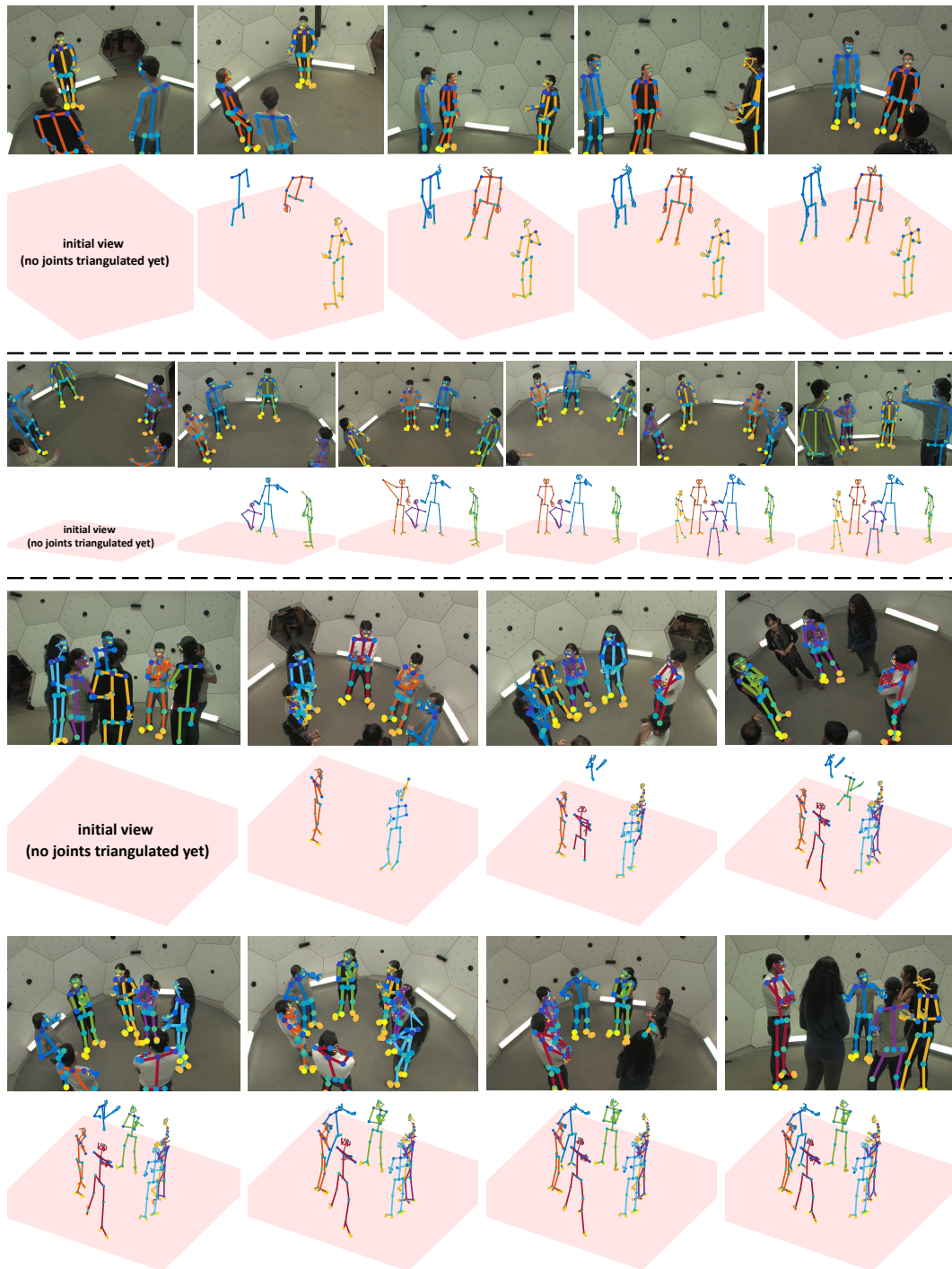

Figure 5: ACTOR operating in three different multi-people scenes (examples delimited by dashed lines). Visualizations are shown for initial active-views and thus have no propagated 3d pose estimates from earlier time steps. Each example shows the views selected by ACTOR, including 2d pose estimates (first view randomly given). Below these we show aggregated 3d pose reconstructions. Top: 3-person scene. One of the persons is reconstructed already at the 2nd view; all of them are reconstructed at the 5th view. The mean 3d reconstruction error decreases from 268 to 51 mm/joint between the 2nd and last view. Middle: 5-person scene, where 3d pose reconstructions improve over the 6 views. The error decreases from 296 to 68 mm/joint. Bottom: 6-person scene, where people stand quite close to each other, which makes it difficult to triangulate all joints due to occlusions. ACTOR observes the scene from 8 diverse views, and the error decreases from 342 to 69 mm/joint.

## Footnotes

[2]Instance-sensitive features generated using a VGG-19 based [25] siamese instance classifier, trained with a contrastive loss to differentiate people on the training set.

[3]The camera history consists of $w$ bins in the azimuth direction and $h$ bins in the elevation direction. It is agent-centered, i.e. relative to the agent's current viewpoint. We set $w = 9$ and $h = 5$.

[4]The precision parameters $m_a$ and $m_e$ are treated as constants, but we anneal them over training as the policy becomes better at predicting camera locations.

[5]In this case we equip ACTOR with an OpenPose system that estimates detailed faces, hands and feet. We do not refine the pre-trained ACTOR model that was trained using the standard OpenPose estimator.

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
