[Supplementary Material]

# Domes to Drones: Self-Supervised Active Triangulation for 3D Human Pose Reconstruction Supplementary Material

**Aleksis Pirinen**[1*], **Erik Gärtner**[1*] **and Cristian Sminchisescu**[1,2]
[1]Department of Mathematics, Faculty of Engineering, Lund University
[2]Google Research
{aleksis.pirinen, erik.gartner, cristian.sminchisescu}@math.lth.se

This supplementary provides more insight into our ACTOR model and experimental setup. Section §1 describes the details of the network architecture, implementation, and hyperparameters. §2 elaborates on how we match 2d pose estimates in space and time using instance features. In §3 we provide 2d reprojection errors onto 2d OpenPose [2] estimates on the Panoptic test splits. Finally, §4 describes further dataset details.

## 1 Model Architecture

See Fig. 1 for a full description of the ACTOR network architecture. ACTOR was implemented in Caffe [5] and MATLAB. We used an open-source TensorFlow implementation of OpenPose[2]. All code and pre-trained weights have been made publicly available.[3]

Figure 1: ACTOR policy architecture. A multi-people 2d pose estimation system (here OpenPose, but any similar deep system would would work) processes an input image. The deep feature maps $B^t$ produced by OpenPose (`conv4_4_CPM`) is fed into the ACTOR policy network and is processed by two convolutional layers with ReLU-activations. The first and second convolutional layers both have $3 \times 3$ kernels with stride 1. Their output dimensions are $8 \times 39 \times 21$ and $4 \times 18 \times 9$, respectively. The max pooling layer has a $2 \times 2$ kernel with stride 2. The output from the second convolutional layer is then concatenated with agent-centric camera rig information about previously visited cameras relative to current position (*Rig*), and auxiliary information such as the number of joints triangulated and number of people detected in the view (*Aux*). The flattened and concatenated data is then fed into three fully-connected layers with tanh-activations with 1024, 512 and 2 neurons respectively. The final output is scaled by two constants to produce radial angles on the viewing sphere.

### 1.1 Hyperparamters

Hyperparameter search was performed using two powerful workstations equipped with several NVIDIA Titan V100 GPU:s. Training a single model for 40k episodes took about 32 hours using one GPU and to speed up results while searching for optimal hyperparameters we trained several model configurations in parallell using Hyperdock [3]. The most important parameters for training ACTOR

---

[*]Denotes equal contribution, order determined by coin flip.
[2]https://gist.github.com/alesolano/b073d8ec9603246f766f9f15d002f4f4
[3]https://github.com/ErikGartner/actor

were learning rate, precision of the the von Mises ($m_a$, $m_e$) and the annealing rate of the precision. See Table 1 for a summary of the values tested for these hyperparameters. In total we trained around 10 different versions of the final model with varying hyperparameters and evaluated each of them on the validation set. Finally, the best model was evaluated on the test dataset and retrained with four additional random seeds to measure the model's sensitivity to the random seed (the model is not very sensitive as indicated in Fig. 2, main paper).

| Hyperparameter | Attempted values |
|---|---|
| Learning Rate | {1e-7, **5e-7**, 1e-6, 5e-6} |
| von Mises precision | {**(1, 10)** → **(25, 50)**, (10, 50) → (20, 100), (10,50) → (100, 500)} |

Table 1: The values tested for the most important hyperparameters when training ACTOR. The final and best values are highlighted in bold. For the von Mises precisions, the arrow indicates linear annealing performed during training (e.g. from $(m_a, m_e) = (1, 10)$ to $(m_a, m_e) = (25, 50)$ for the best configuration).

## 2 Matching Multiple People

ACTOR reconstructs multiple people in both space and time from 2d pose estimates. In order to track and match these estimates we compute instance sensitive features. These deep features can then be stably matched to each other using the Hungarian algorithm and the L2 distance to compute the matching cost.

We trained an instance classifier structured as a siamese network [1] using a contrastive loss [4] that aims to produce 50-dimensional features for each person that can be used to distinguish individuals. As input the instance classifier takes VGG-19 [7] features from the bounding box of the 2d pose estimate. The instance classifier is trained for 40k iterations on the training set with a mini-batch size of 16 where half contains positive pairs and the other half contains negative pairs. The training examples are sampled randomly in both space and time yielding a robust classifier. Lastly, the instance classifier is fine-tuned for 2k iterations on each scene, creating scene-specific versions of the classifier that are slightly adapted to the environment of those scenes. This tuning is performed outside the range of the active-sequence in which the agent operates.

At the start of an active-sequence the agents is given an appearance model for each target it should reconstruct. These appearance models are averages of $K$ different instance features computed for each target in the scene but from time-freezes that are *not part of the current active-sequence*. We denote the $i$:th instance feature for the $l$:th person by $\boldsymbol{u}_i^l$, with $i = 1, \ldots, K$. In practice we use $K = 10$. Then we set as appearance model $\boldsymbol{m}^l$ to:

$$\boldsymbol{m}^l = \text{median}(\boldsymbol{u}_1^l, \ldots, \boldsymbol{u}_K^l) \tag{1}$$

For each camera location we compute the distance between the instance features of each detected person to all appearance models in that scene. This gives us a cost matrix whose elements are $c^{j,l} = \|\boldsymbol{u}^j - \boldsymbol{m}^l\|_2^2$, i.e., the cost to match detection $j$ to person $l$. Given this matrix we assign detections according to the Hungarian algorithm. Since there might be false detections by the 2d pose estimator and not all people are visible from every camera location we filter out matches with a cost larger than a threshold $\mathcal{C}$, such that all matches $c^{j,l} \leq \mathcal{C} = 0.5$.

If a person is never detected in an active-view, and if it does not have a previous temporal backup to use as 3d pose reconstruction (cf. §3 and the implementation details in §4 of the main paper), we set each joint estimate to the ground-truth center hip location. Obviously, this estimate is implausible and highly inaccurate – it is used only to compute average errors (not including such an estimate when computing average errors would be another option, but this would not penalize missed persons).

## 3 Reprojection Errors onto OpenPose 2d Estimates

The 3d ground-truth in Panoptic is generated from exhaustive triangulation of 2d pose estimates [6], but those 2d pose estimates are not from OpenPose. Thus it is relevant to also look at reprojection

Figure 2: Mean 2d reprojection errors per joint relative to OpenPose 2d estimates vs number of cameras on the test sets. Left: Multi-people data. Right: Single-people data. ACTOR reduces the 2d reprojection error faster than the heuristic baselines, particularly for multi-people data. Single-person scenes are easier to reconstruct, especially when using many cameras – also note that all models converge close to the error of the oracle in this case.

errors onto the OpenPose 2d estimates, since these errors are not affected by any potential incorrect bias in the 3d ground-truth. Such reprojection errors are shown in Fig. 2. We note that ACTOR is more accurate relative to the oracle in this metric. For single-people data the agent even converges close to the oracle, while the oracle is still slightly better for multi-people data due to its more difficult nature with occlusions. ACTOR yields lower reprojection errors than the heuristic baselines, with an exception at 2 cameras for multi-people data where *Max-Azim* is more accurate. Note however that ACTOR was not trained to produce accurate estimates at any fixed number of cameras, but rather to quickly triangulate all joints. Despite this, we outperform the baselines in the vast majority of cases.

## 4 Additional Dataset Insights

Table 2 shows the size and split of the Panoptic dataset [6] into train, validation and test sets. The data was created using scripts that downsampled from 30 FPS to 2 FPS to increase movement between frames.

|  | Train | Val | Test | All |
|---|---|---|---|---|
| **Mafia** | 53,100 | 27,900 | 33,728 | 114,728 |
| **Ultimatum** | 27,960 | 4,340 | 55,825 | 88,125 |
| **Pose** | 51,079 | 29,672 | 59,288 | 140,039 |
| **All** | 132,139 | 61,912 | 148,841 | 342,892 |

Table 2: The number of images in our dataset categorized by scene type and subset type (training, validation, and testing). Note that *Mafia* and *Ultimatum* are complex multi-people scenes and that they account for more than half of the dataset.