[Reviews · NeurIPS 2019]

Reviewer 1



I really like this paper primarily for its novel technology idea to select the best camera views for 3D pose estimation in multi-person scenario with occlusions. I can see similar idea can be used in many other relevant problems when it needs to find the best subset of camera views, like real-time holoportation. Even though the technology itself is not super exciting and completely new, applying RL to tailor this specific problem is novel enough for a paper. I have a few minor questions * what's the trends of performance when the # of cameras > 10? from the plots, it seems that the gain trends to be small when using more cameras compared with other baselines, can you explain why? for example, the performance is almost close to Max-Azim when the # of camera is 10(see figure 2) * I believe there is a strong reason, but I am curious why not using the exhaustive triangulation of 2d pose estimate in practical applications? What's the benefits of using this auto-selection based algorithms. * This paper has not discussed any downside of this approach, to be fair, what are the things need to be careful when applied for other problems like Multiview 3D reconstruction? *other minor issues (1) Figure 5 is missing (2) line 223~224, the sentence " The corresponding results but for 2d reprojection errors onto the OpenPose 2d estimates are given in Fig. 3. Looking also at reprojection errors is relevant" is very hard to understand, please polish it. ***Rebuttal***: I've read the authors rebuttal and other reviewers' feedback, I keep my original rating. I like the idea and problem formulation, however, as pointed out by reviewers as well, the practical impact of this work for real application is not quite clear. I am okay to accept, but will not fight if it is rejected.

Reviewer 2



This paper focuses on the problem of 3D human pose reconstruction from multiple viewpoints in video sequences. The system assumes that a set of 2D human poses are available (e.g. by using OpenPose), coming from different camera viewpoints, and the paper proposes a pipeline for simultaneously selecting the next viewpoint that would be needed to decrease the reconstruction error and generating the 3D reconstruction. An artificial agent, named ACTOR, is defined for the purpose of selecting the camera viewpoints, following a reward-based approach. The proposed system is evaluated on the CMU Panoptic dataset, which contains 480 VGA camera views. The experimental results show that by using the proposed agent the reconstruction error is better than using the baseline methods. Although worse than selecting the cameras in a greedy fashion ("Oracle" case). * Originality: the originality comes from the definition of the agent that selects the most relevant cameras for the triangulation stage. The remaining components of the system do not seem to be novel. * Quality: the technical content of the paper appears to be correct. No flaws have been found in the experimental setup. - From Table 1, it does not seem that the difference between ACTOR cases and Random and Max-Azim is large, and they could be probably faster. Therefore, how much time is needed to select a camera viewpoint at each time stamp compared to the baseline methods? * Clarity: the paper is well-written. Technical and implementation details are available. However, the description of the dataset where the system is evaluated is minimal. In fact, I do not see clearly stated if the almost 500 available cameras of the Panoptic studio are used for the experiments. * Significance: the idea of automatically selecting the most suitable cameras for the triangulation purpose is interesting. However, the practical situation where it is validated is quite rare, i.e., it is very unusual having such a large amount of cameras available. This limits the range of applications where this can be applied. Basically, if only few cameras were available, there would be no need of selection, all of them might be used. *** Post-rebuttal *** I have carefully read the answer of the authors and my overall score goes towards its acceptance.

Reviewer 3



# Originality The task is new, though there are related works on autonomous reconstruction of 3D scenes or next-best-view selection for object pose estimation. There is a lack of thorough discussion on these related tasks and the special challenge in the setting of 3D human pose reconstruction, as well as how the special challenge is solved in this work. # Quality It is a solid in general. The proposed approach is technically sound. The experiments and baselines are well designed and sufficiently demonstrate the effectiveness of the proposed approach. One concern is that the reward is based on visibility of body joints in the selected views, and the authors claim that no annotation is needed for training. The question is how to know the true visibility if there is no annotation? # Clarity The idea and methodology are clearly presented. The paper is well organized and easy to follow. # Significance This is a new task. My main concern is if this task is useful in practice. For the multi-view setting, I was not able to imagine why not using all views. For the active observer setting, the assumption that human is static is impractical. Pose estimation is usually for motion capture where people are moving.

[Author Response · NeurIPS 2019]

We thank the reviewers for their valuable feedback and note that all reviewers agree about the novelty of our work.

**AR1**: *Results for more than 10 cameras:* Triangulation converges when adding more cameras, thus narrowing ACTOR's
improvement over the baselines. Using many cameras however requires significant computation, hence we are interested
in finding small yet informative subsets of cameras that yield good results. We chose to run the model for 1-10 cameras
as evaluating the Oracle is slow for large camera sets, cf. Fig. 3 right. Results for 15 cameras on multi-people data:
ACTOR = 69.99, Random = 74.71, Max-Azim = 73.79, Oracle = 57.24 (mm/joint hip-translated). Compared to
running for 10 cameras, ACTOR keeps its gain over Max-Azim and significantly closes in on the Oracle, see Table 1.

*Minor issues:* (1) Fig. 5 is not missing; see page 8 and line 240. (2) We will clarify line $\sim$ 223-224 in the paper.

**AR2**: *Runtimes and baseline comparisons:* Fig. 3-4 right show runtimes; OpenPose adds 0.134 s/image and our policy
adds 0.005 s/image (25x faster), see line 237-240. ACTOR is almost as fast as the baselines while *significantly more*
*accurate* when considering fewer views, e.g. 94 mm/joint (34 %) better than Max-Azim at 3 views (Table 1), yet only
6% slower (Fig. 3 right). When using more cameras triangulation converges, decreasing the need for intelligent camera
selection. The Oracle cheats by using 3d ground-truth when selecting views. It is greedy, as exhaustively evaluating the
exponential set of views is infeasible. It is still very slow since it evaluates all cameras before selecting the next view.

*Dataset description:* We use the HD cameras, of which there are about 30 per scene. The HD cameras are used since
they provide better image quality than VGA and are sufficiently dense, yet spread apart far enough to make each
viewpoint unique. We will include this in an extended dataset description for the camera-ready.

*More realistic setup:* To mimic a single camera setup, we experimented on multi-people data assuming time elapses
0.5 s between *every* selected view. Note that some joints which move drastically between views cannot be triangulated.
Therefore we extend ACTOR with a monocular 3d pose estimator (DMHS, Popa et al., CVPR, 2017) as a fallback.
*Without refining the policy* our results (in auto-mode) are: ACTOR = 123.3, Random = 138.7, Max-Azim = 150.1,
Oracle = 106.1 (mm/joint hip-translated). ACTOR outperforms the baselines also in this setting (cf. Table 1). While
these are only proof-of-concept results they clearly indicate the potential of our model for single camera setups.

*State-of-the-art:* Our results are significantly better than state-of-the-art 3d pose estimators such as MubyNet (Zanfir et
al., NeurIPS, 2018). Their errors are about 150 mm/joint on multi-people data, while ACTOR obtains 96 mm/joint in
auto-mode, see Table 1. There exists no prior work on active 3d human pose estimation, so results are hard to compare
as they are framed in completely different settings (monocular 3d estimation vs active 2d-to-3d triangulation).

**AR3**: *No annotations needed for training:* We emphasize that our reward is *not* based on ground-truth visibility – it
relies solely on the automatic 2d pose (body joint) detection. Thus our approach requires no annotations in training.

*Next-best-view (NBV) and object pose estimation:* NBV is a very general concept, and our formulation can be seen as
one way of modeling and implementing it in the framework of deep reinforcement learning. We further address the
novel problem of intelligent view selection for 3d human pose reconstruction (or generally articulated structures), which
is more challenging than rigid 3d reconstruction and pose estimation or object detection[1]: i) Dimensionality is much
higher than object detection and the problem is more complex than rigid 3d reconstruction - humans are articulated
and deformable, scenes contain multiple people; ii) Realistic training data for supervised 3d human pose estimation is
scarce, and our proposed methodology requires *no annotations* and instead uses self-supervision to learn the policy;
iii) Reconstruction difficulty is impacted by occlusions and the pose of targets, hence our model adaptively selects a
*variable number of views* depending on scene complexity; iv) NBV for e.g. object pose estimation is formulated such
that, at each step, one selects views by assessing a utility function over a set of candidates. Most design choices are
handcrafted, decisions are local. We frame the task as a deep RL problem where the policy is trained to maximize a
*global objective*, searching over entire sequences of viewpoints, and by triangulating as many joints as possible.

**AR2, AR3**: *Practical applications:* We use our setup as a proxy for a moving observer and to develop viewpoint
selection strategies for e.g. intelligent holoportation (AR1) while providing a framework for *reproducible* experiments.
Practical developments of our methodology would include e.g. real-time intelligent processing of multi-camera
(Panoptic/Lightstage) video feeds or control policies for a drone observer. In the latter case the model would further
benefit from being extended to account for physical constraints, e.g. a single camera and limited speed. Our paper is a
key first step since it presents fundamental methodology required for future applied research.

**AR1, AR3**: *Why not exhaustive triangulation:* Using all views requires significant computation, even with one of the
fastest pose estimators such as OpenPose (0.134 s/image vs only 0.005 s/image for our policy, see line 238-239). Thus
exhaustive computation does not suit real-time scenarios (processing 30 cameras takes over 4 s / frame). Also, assuming
a physical observer (e.g. a drone), the need for a view selection strategy is crucial since covering all views is infeasible.

## Footnotes

[1]Obviously there is no one size fits all – details on the output representation, level of detail, single versus multiple structures, occlusions, observation setup, or desired reward would affect formulation and modeling choices for each problem.


[Meta-Review · NeurIPS 2019]

This paper presents a method based on agent to select the best view for a 3D pose estimation given 2D poses available. The proposed system is evaluated on the CMU Panoptic dataset, which contains 480 VGA camera views. The reviewers agree with the originaly of the task and the proposal and on the clearness of the presentation Rebuttal was convincing and thus also the area chair agrees for an acceptance of the paper.